# Effects of the NF-κB Signaling Pathway Inhibitor BAY11-7082 in the Replication of ASFV

**DOI:** 10.3390/v14020297

**Published:** 2022-01-31

**Authors:** Qi Gao, Yunlong Yang, Yongzhi Feng, Weipeng Quan, Yizhuo Luo, Heng Wang, Jiachen Zheng, Xiongnan Chen, Zhao Huang, Xiaojun Chen, Runda Xu, Guihong Zhang, Lang Gong

**Affiliations:** 1Key Laboratory of Zoonosis Prevention and Control of Guangdong Province, College of Veterinary Medicine, South China Agricultural University, Guangzhou 510462, China; qigao2021@scau.edu.cn (Q.G.); yunlongyang@stu.scau.edu.cn (Y.Y.); fyz@stu.scau.edu.cn (Y.F.); weipengquan@stu.scau.edu.cn (W.Q.); lawzz@stu.scau.edu.cn (Y.L.); wangheng2009@scau.edu.cn (H.W.); zhengjc@stu.scau.edu.cn (J.Z.); cxn201314@stu.scau.edu.cn (X.C.); yingwenmulu@stu.scau.edu.cn (Z.H.); xiaojunchen@stu.scau.edu.cn (X.C.); 18819255305@163.com (R.X.); 2Research Center for African Swine Fever Prevention and Control, South China Agricultural University, Guangzhou 510642, China; 3Maoming Branch, Guangdong Laboratory for Lingnan Modern Agriculture, Maoming 525000, China; 4African Swine Fever Regional Laboratory of China (Guangzhou), Guangzhou 510642, China

**Keywords:** African swine fever virus, IL-1β, IL-8, NF-κB signaling pathway, BAY11-7082

## Abstract

African swine fever virus (ASFV) mainly infects the monocyte/macrophage lineage of pigs and regulates the production of cytokines that influence host immune responses. Several studies have reported changes in cytokine production after infection with ASFV, but the regulatory mechanisms have not yet been elucidated. Therefore, the aim of this study was to examine the immune response mechanism of ASFV using transcriptomic and proteomic analyses. Through multi-omics joint analysis, it was found that ASFV infection regulates the expression of the host NF-B signal pathway and related cytokines. Additionally, changes in the NF-κB signaling pathway and IL-1β and IL-8 expression in porcine alveolar macrophages (PAMs) infected with ASFV were examined. Results show that ASFV infection activates the NF-κB signaling pathway and up-regulates the expression of IL-1β and IL-8. The NF-κB inhibitor BAY11-7082 inhibited the expression profiles of phospho-NF-κB p65, p-IκB, and MyD88 proteins, and inhibited ASFV-induced NF-κB signaling pathway activation. Additionally, the results show that the NF-κB inhibitor BAY11-7082 can inhibit the replication of ASFV and can inhibit IL-1β and, IL-8 expression. Overall, the findings of this study indicate that ASFV infection activates the NF-κB signaling pathway and up-regulates the expression of IL-1β and IL-8, and inhibits the replication of ASFV by inhibiting the NF-κB signaling pathway and interleukin-1 beta and interleukin-8 production. These findings not only provide new insights into the molecular mechanism of the association between the NF-κB signaling pathway and ASFV infection, but also indicate that the NF-κB signaling pathway is a potential immunomodulatory pathway that controls ASF.

## 1. Introduction

African swine fever (ASF) is an acute and highly infectious disease of pigs caused by African swine fever virus (ASFV), and is transmitted by domestic pigs, wild boars, and insect vectors [1]. ASF originated from Africa and was first reported in 1921 in Kenya, but the virus has spread to Europe, South America, and Asia. The first ASFV case in China was reported on August 3, 2018, and it subsequently spread throughout the country within a few months. Presently, there are 24 genotypes and eight serogroups of ASFV [2], and the main strains in China are genotype II and serogroup 8 [3]. ASFV is highly restricted to porcine cells of the monocyte/macrophage lineage and preferentially infects porcine alveolar macrophages (PAMs) [4]. Acute infection of pigs with ASFV can result in up to 100% mortality. Currently, there are no commercially approved vaccines or antiviral drugs to control the disease, so control measures rely on culling infected animals, restricting animal movement, and biosecurity prevention and control [5,6]. Highly virulent isolates of ASFV often cause a peracute to acute disease progression with high fever (>41 °C) and a range of clinical signs, including anorexia and lethargy, which occur within a few days of infection [7]. The genome length of ASFV ranges from 170 to 190 kb [8]. The genome of ASFV encodes nearly 200 proteins, of which more than 50 structural proteins are packaged into virus particles, which play an important role in the infection stage of the virus and participate in the process of genome replication and viral infection [9].

Transcriptomics analysis is an advantageous method to understand the relationship between genotype and phenotype, and to gain a deeper understanding of the potential pathways and mechanisms of cell fate, development, and disease progression [10]. Various bioinformatics analyses based on high-throughput proteomics data can identify the functional relationship between different proteins from a whole perspective [11]. A comprehensive understanding of the interaction between the virus and the host requires the research and analysis of the molecular complexity and multiple levels of variation in the genome, epigenome, transcriptome, proteome, and metabolome [12]. Therefore, this study conducted an in-depth study of the interaction mechanism between ASFV and the host through the analysis of multiple omics data and experimental phenomena. To better understand molecular mechanisms mediating virus pathogenesis and immune evasion, we used transcriptome and proteomics analysis to identify gene expression changes after ASFV infection in PAMs.

Immunosuppression caused by ASFV is mainly associated with monocytes and macrophages and altered expression patterns of cytokines in blood and tissues [13]. Altered patterns of cytokine expression were observed in piglets infected with ASFV. There was an overexpression of CXCL4, CXCL8, CXCL10, IFN-α, TNF-α, IL12p40, IFN-β, IL-18, IL-1β, and IL-1α, and a decrease in the expression of MHC I and CD14/CD16 at the mRNA level in macrophages after ASFV infection [14,15,16]. IL-1 and IL-8 belong to the cytokine families and are associated with humoral immune responses, and their expression profiles can reflect the immune status of an animal [17]. Previous studies have shown that monocyte and macrophage lineage cells are the main targets of ASFV [18,19]. Therefore, it is important to examine the effect of ASFV infection the expression profiles of cytokines in monocytes and macrophage lineage cells.

Nuclear factor-κB (NF-κB) is a transcription factor that regulates genes related to immunity, inflammation, and cell activity [20]. For a long time, the NF-κB pathway has been regarded as a typical pro-inflammatory signal transduction pathway. During injury or infections, tissues rapidly release pro-inflammatory factors, such as interleukin 1 (IL-1) and tumor necrosis factor α (TNF-α). IL-1 and TNF-α can activate the NF-κB signaling pathway, which can promote the secretion of inflammatory factors, chemokines, and adhesion molecules [21,22,23]. Additionally, the NF-κB pathway is involved in inflammatory diseases, and several studies have focused on the development of anti-inflammatory drugs for regulating NF-κB [24,25,26]. However, inflammation is a complex physiological process, and inflammatory response is characterized by the regulation of the secretion of pro-inflammatory and anti-inflammatory factors and the activity of various signal transduction pathways. Moreover, although cytokine production is affected in pigs infected with ASFV, the underlying regulatory mechanisms remain largely unknown.

Therefore, the purpose of this study is to use transcriptomic and proteomic to analyze the changes in the NF-κB signaling pathway and the expression of cytokines IL-1β and IL-8 after PAMs infected by ASFV, as well as the NK-κB signaling pathway in ASFV. Furthermore, we used an NF-κB inhibitor, BAY11-7082 [27], to block NF-κB activation to knockdown MyD88 protein levels in porcine alveolar macrophages (PAMs) to understand the role of NF-κB signaling pathways in PAMs infected with ASFV. Studying the mechanism of the NF-κB signaling pathway in the infection process provides a theoretical basis for elucidating the pathogenic mechanism of ASFV.

## 2. Materials and Methods

### 2.1. Cell Culture and Virus

Primary porcine alveolar macrophages (PAMs) were collected from 20–30 day-old SPF (Specific Pathogen Free, SPF) pigs. Primary porcine alveolar macrophages (PAMs) were maintained in 10% FBS RPMI 1640 medium (Gibco, Waltham, MA, USA) supplemented with 2 mM l-glutamine, 100 U/mL gentamycin, and 0.4 mM nonessential amino acids at 37 °C in a 5% CO_2_ atmosphere saturated with water vapor. Viral titers were determined as the amount of virus causing hemadsorption (for HAD isolates) in 50% of infected cultures (HAD_50_/mL). Samples with mycoplasma contamination were identified and excluded using a Mycoplasma stain detection kit (Beyotime, Shanghai, China). The high virulence, hemadsorbing ASFV isolate GZ201801 (GenBank: MT496893.1) was isolated in Guangzhou, China, is p72 genotype II, and is preserved in the Infectious Diseases Laboratory of South China Agricultural University.

### 2.2. Reagents and Antibodies

NF-κB inhibitor BAY11-7082 (Sclleck, Shanghai, China, S2913), and α-Tubulin rabbit polyclonal antibody (Beyotime, Shanghai, China, AF0001). Rabbit monoclonal antibodies against NF-κB p65 (D14E12) XP, phospho-NF-κB p65 (Ser536; 93H1), IκBα (L35A5), phospho-IκBα (Ser32; 14D4), and MyD88 (D80F5) were purchased from Cell Signaling Technology (Danvers, MA, USA). IRDye^®^ 800CW goat anti-rabbit IgG antibody and goat anti-mouse IgG antibody (highly cross adsorbed) were purchased from LI-COR Biosciences (Lincoln, NE, USA). The p30 antibody is a murine monoclonal antibody prepared by our laboratory and used in both Western Blot and IFA experiments.

### 2.3. HAD Assay

As previously described [28], primary PAMs were cultured in 96-well plates and infected with 10-fold diluted ASFV (GZ201801). The quantity of ASFV was determined by identification of characteristic rosette formation representing hemadsorption of erythrocytes around infected cells. Cultures were observed for HAD phenomena over 7 days, and HAD_50_ was calculated using the method of Reed and Muench [29]. Primary PAMs were infected with GZ201801-ASFV at an MOI of 1.

### 2.4. Immunofluorescence Assay

Primary PAMs infected with ASFV at an MOI of 1 were seeded on a 24-well plate and incubated with anti-ASFV p30 protein monoclonal antibody, which was previously diluted with 2% bovine serum albumin (BSA) at a ratio of 1:500. At 24 h post-infection, the cells were washed five times with PBS (1 mL each time), fixed in 500 μL 3.7% paraformaldehyde for 30 min at room temperature, permeabilized in 1 mL 0.1% (*w*/*v*) Triton-100 for 20 min at room temperature, and then incubated in the dark with a secondary antibody diluted with 2% BSA (1:200) for 1 h at 37 °C in a humid chamber. Thereafter, cell nuclei were stained with 4′,6-diamino-2-phenylindole (DAPI) at room temperature for 5 min, and then washed three times with PBS. Cell fluorescence was observed using an immunofluorescence microscope.

### 2.5. RNA Isolation, cDNA Library Preparation, and Sequencing

PAM cells were infected with ASFV (MOI 1) or blanks (control) and harvested at 3 h, 12 h, and 48 h post-infection. Total RNA was extracted from the cells using RNAiso Plus (TAKARA, 9108), according to the manufacturer’s instructions. RNA quantity and purity were assessed using a Thermo NanoDrop Lite spectrophotometer (ND-NDL-US-CAN; Thermo Fisher Scientific, Waltham, MA, USA). Total RNA was processed by the mRNA enrichment method: mRNA with polyA tail was enriched with magnetic beads with OligodT. The obtained RNA was fragmented by interrupting buffer, reverse transcribed with random N6 primers, and cDNA double-stranded was synthesized to form double-stranded DNA. The end of the synthesized double-stranded DNA is blunted and phosphorylated at the 5′ end, forming a sticky end with an “A” protruding from the 3′ end, and then ligated with a bubbling linker with a protruding “T” at the 3′ end. The ligated products were amplified by PCR with specific primers. The PCR product was heat-denatured into single-stranded DNA, and then a single-stranded circular DNA library was obtained by circularizing the single-stranded DNA with a bridge primer. The constructed library was checked for quality and sequenced after it is qualified. This project used the DNBSEQ platform to sequence the samples, and each sample produced an average of 6.68 G of data. The average alignment rate of the sample alignment genome was 90.86%, and the average alignment rate of the aligned gene set was 66.68%; a total of 17,158 genes were detected.

### 2.6. Data Analysis of RNA-Seq

This project used the filtering software SOAPnuke independently developed by BGI for filtering to remove reads containing joints (joint contamination), remove reads with an unknown base N content greater than 5%, and remove bases with a mass value of less than 15, which account for the total base of the reads of which the proportion of cardinality is greater than 20% of low-quality reads. The raw “reads” were filtered to obtain high quality de novo transcriptome sequence data. First, all reads with adaptor contamination were discarded. Second, reads with unknown nucleotides comprising more than 5% were removed. Third, low-quality reads with ambiguous sequence “N” were discarded. After quality control, clean reads were aligned with the reference sequence GCF_000003025.6_Sscrofa11.1(NCBI). Thereafter, differential expression analysis was performed to determined differentially expressed genes (DEGs) in each treatment group. Unigenes with fold change > 2 and Q value ≤ 0.05 were considered significantly differentially expressed. Using BGI’s Dr. Tom multi-omics data mining system, clustering heat map, Venn, GO, and KEGG analysis and results display of differential genes and proteins were performed. Functional annotation and pathway analysis of DEGs were performed using the gene ontology (GO) and Kyoto Encyclopedia of Genes and Genome (KEGG) databases and the resulting graphs were presented. A protein interaction network was also constructed.

### 2.7. qPCR

DNA was isolated using the Axyprep Body Fluid Viral DNA/RNA Miniprep Kit (Axygen, China, AP-MN-BF-VNA). A total of 1 μL of DNA was used for real-time PCR assay using AceQ Universal U+ Probe Master Mix V2 (Vazyme, Nanjing, China, Q711-02). The relative quantity of viral DNA was determined using the CADC p72 primers and a probe experiment. The gene-specific primer and probe sequences are listed in Table 1. Total RNA was isolated using RNAiso Plus (Takara, 9108), and reverse transcribed into cDNA using the HiScript II 1st Strand cDNA Synthesis Kit (+gDNA wiper) (Vazyme, China, R212-02). A total of 1 μL of cDNA was used for real-time PCR assay using ChamQ Universal SYBR qPCR Master Mix (Vazyme, China, Q711-02). The relative quantity of cell RNA was determined by performing a comparative Ct (ΔΔCt) experiment using GAPDH as an endogenous control. Gene-specific primer sequences were designed using the Oligo7 software (Table 1).

### 2.8. Western Blot Analysis

For western blot analysis, cells were lysed in RIPA buffer (Beyotime, Shanghai, China) and denatured by adding 4× Laemmle SDS-PAGE buffer (containing DL-dithiothreitol), followed by heating for 15 min at 100 °C. The proteins were then separated on SDS-PAGE gels and transferred onto nitrocellulose membranes using a Trans-Blot Turbo rapid transfer system (Bio-Rad, Hercules, CA, USA), according to the manufacturer’s instructions. The membranes were blocked in 5% defatted milk (dissolved in Tris-buffered saline (TBS)) for 1 h at 37 °C and then incubated with a primary antibody for 1 h at room temperature or overnight at 4 °C. The membranes were then washed three times (5 min per time) using a wash buffer (TBS containing 0.1% Tween 20) and incubated with an IRDye^®^ 800CW secondary antibody for 1 h at 37 °C. The membranes were washed three times in wash buffer and imaged using an Odyssey Imaging System (LI-COR, USA) to visualize the protein bands. α-Tubulin was used as the loading control.

### 2.9. Cell Viability Assay

PAM cells were seeded in a 96-well plate, containing maintenance medium and different concentrations of BAY11-7082, and incubated at 37 °C for 48 h. DMSO was used as a control (vehicle). Thereafter, 10 μL of CCK-8 solution (Beyotime, C0039) was added and the incubation was continued for 1 h. After incubation, the absorbance was read at 450 nm using an enzyme-labeled instrument and cell viability was calculated.

### 2.10. Statistical Analysis

Pathway analysis and functional annotation of DEGs, and differentially expressed proteins identified by transcriptomics and proteomics were performed using the KEGG, GO, and KOG databases, respectively. The STRING method was used for protein network interaction analysis. The SPSS software package (SPSS for Windows version 13.0, SPSS Inc., Chicago, IL, USA) was used to perform statistical analysis of data obtained during the experiment. The difference between the experimental group and the control group was analyzed by one-way ANOVA using GraphPad Prism 8 (GraphPad Software, San Diego, CA, USA). Values are expressed in graph bars as the mean ± standard deviation (SD) of at least three independent experiments. Statistical significance was set at * *p* < 0.05, *** *p* < 0.01, and **** *p* < 0.001.

## 3. Results

### 3.1. Proliferation and Growth Curve of ASFV In Vitro

To assess the viral growth dynamics, primary PAMs were infected at an MOI of 1, and the cell supernatants and cell mixed liquid were collected at different time points post-infection for viral genome quantification by qPCR. The titer for the first passage stock was 10^6.4^ HAD50/mL (Figure 1).

### 3.2. Conjoint Analysis of Transcriptomics and Proteomics

To explore how ASFV infection regulates host gene expression, transcriptomic and proteomic analyses of ASFV-infected cells were performed. Briefly, PAMs permissive for ASFV were infected with ASFV at a multiplicity of infection (MOI) of 1. Cells were harvested at 3 h, 12 h, and 48 h post-infection (p.i.), and total RNA was extracted and subjected to genomic analysis. RNA and protein samples isolated from living cells without any treatment were subjected to RNA and amino acid sequencing (Figure 2A,B). Compared with the control group, 1018, 3752, and 5131 DEGs were identified in the 3 h, 12 h, and 48 h ASFV-infection groups, respectively. Additionally, 410 DEGs were common to the 3 h, 12 h, and 48 h ASFV-infection groups (Figure 2C). Regarding differentially expressed proteins, compared with the control group, 118, 324, and 2143 differentially expressed proteins were identified in the 3 h, 12 h, and 48 h ASFV-infection groups, respectively; additionally, the three groups had 37 differentially expressed proteins in common (Figure 2D). KEGG pathway enrichment analysis of the DEGs showed that DEGs identified in the 3 h and 12 h infection groups were mainly enriched in the cytokine–cytokine receptor interaction, C-type lectin receptor signaling pathway, TNF signaling pathway, IL-17 signaling pathway, Jak-STAT signaling pathway, toll-like receptor signaling pathway, NF-kappa B signaling pathway, and chemokine signaling pathway (Figure 3A,B), whereas DEGs identified in the 48 h infection group were mainly enriched in cell cycle, Th17 cell differentiation, the TNF signaling pathway, apoptosis, Th1 and Th2 cell differentiation, the C-type lectin receptor signaling pathway, osteoclast differentiation, spliceosome, and metabolic pathways (Figure 3C). KEGG pathway enrichment analysis of differentially expressed proteins in the 3 h and 12 h infection groups revealed that they were mainly enriched in the TNF, NF-kappa B, IL-17, NOD-like receptor, chemokine, and toll-like receptor (Figure 3D,E), whereas differentially expressed proteins identified in the 48 h infection group were mainly enriched in the Fc epsilon RI signaling pathway, purine metabolism, insulin signaling pathway, C-type lectin receptor signaling pathway, cell apoptosis, and Hippo signaling pathway (Figure 3F).

### 3.3. ASFV Early Infection Activates NF-κB Signaling Pathway

According to the multi-omics joint analysis, DEGs in the 3 h and 12 h infection groups were mainly enriched in the NF-κB signaling pathway. Protein network interaction analysis of the differentially expressed genes revealed that IL-1β, IL-8, and NF-κB were located in the center of the network and were related to most of the DEGs (Figure 4A). Venn diagram analysis showed that among the common DGEs in the 3 h and 12 h infection groups compared with the control group, 11 differential genes were enriched in the NF-κB signaling pathway, including CXCL8, IL-1β, CCL4, IκBα, TNF, NF-κB, CFLAR, PTGS2, TRAF1, PLAU, and IL-1β2 (Figure 4B). The mRNA levels of IL-1β, IL-1β2, IL-8, NF-κB and IκBα were up-regulated with increasing infection time (shown in red font in Figure 4C). The expression of viral p30 protein in PAMs increased with an increase in the ASFV infection time (Figure 5). When the NF-κB signaling pathway is activated, both NF-κB and IκB proteins with a dimer structure undergo phosphorylation modification, and the phosphorylated IκB (PIκB) is degraded, resulting in the phosphorylated NF-κB (PP65) entering the nucleus and causing an immune response. Additionally, myeloiddifferentiationfactor88 (MyD88) is a key linker molecule that activates the NF-κB signaling pathway. The results show that the levels of phospho-NF-κB p65, p-IκB, and MyD88 proteins increased with infection time from 3 to 12 h post-infection and decreased after 48 h post-infection (Figure 5), indicating that the MyD88/NF-κB signaling pathway is activated early in ASFV infection.

### 3.4. BAY11-7082 Inhibits the Overexpression of Phospho-NF-κB p65 Caused by ASFV Infection

Western blotting was used to measure the expression of the phospho-NF-κB p65 protein in each group of PAMs at 0, 3, 6, and 12 h. Compared with the control, there was a significant increase in the phospho-NF-κB p65 level after ASFV infection; however, this effect was reversed by the NF-κB inhibitor (BAY11-7082), this phenomenon shows that BAY11-7082 inhibited the levels of phospho-NF-κB p65; detection of cell viability of the inhibitor BAY11-7082 (Appendix A). The additional time of BAY11-7082 is the same as the time of ASFV infection with PAMs. Tubulin expression was used as an internal reference protein (Figure 6).

### 3.5. BAY11-7082 Inhibits the Overexpression of p-IκB Caused by ASFV Infection

Compared with PAMs incubated with ASFV only, there was a significant decrease in the p-IκB protein level in PAMs co-incubated with ASFV and BAY11-7082 for 3, 6, and 12 h. Western blotting was used to measure the expression of the p-IκB protein at 3, 6, and 12 h in each group of PAMs. Compared with the control group, there was an increase in the p-IκB protein level after ASFV infection; however, this effect was reversed by the NF-κB inhibitor (BAY11-7082). Tubulin expression was used as a positive control (Figure 7).

### 3.6. BAY11-7082 Inhibits the Overexpression of IL-1β and IL-8 Caused by ASFV Infection

Furthermore, the mRNA expression profiles of IL-1β and IL-8 in the PAMs after ASFV infection was examined by real-time PCR (RT-PCR) to determine whether IL-1β and IL-8 were activated by ASFV infection. There was a significant increase in IL-1β and IL-8 mRNA expression levels of PAMs incubated with ASFV only with an increase in infection time from 3 to 12 h. In the experimental group, the NF-κB inhibitor (BAY11-7082) was added at the same time as when PAMs cells were inoculated with ASFV. Compared with PAMs incubated with ASFV alone, co-incubation with the NF-κB inhibitor (BAY11-7082) significantly inhibited IL-1β (Figure 8A) and IL-8(Figure 8B) mRNA expression levels at 3, 6, and 12 h. These results are expected since the expression of these genes is upregulated by NF-kB.

### 3.7. BAY11-7082 Inhibits ASFV Replication

Furthermore, the ASFV replication inhibitory effects of BAY11-7082 were examined by real-time PCR. Compared with the 12 h infection ASFV group, there was a significant decrease in the copies of the ASFV B646L gene after treatment with 10μM BAY11-7082 (Figure 9A). At the same time, a change of virus titer after BAY11-7082 treatment was detected, and it was found that the virus titer decreased significantly (Appendix A). The early viral structural protein p30, which is encoded by the CP204L gene, was detected as early as 2–4 h p.i. and throughout the viral replication cycle. Therefore, we examined the expression of p30 in primary PAMs infected with virus stock. To determine whether ASFV proliferation was inhibited by BAY11-7082, ASFV p30 protein expression was determined by Western blotting and immunofluorescence assay. There was an increase in the expression of the p30 protein in the ASFV group with an increase in infection time from 3 to 12 h. There was a significant decrease in the expression of the ASFV p30 protein in the BAY11-7082 group compared with that of the ASFV group at 3, 6, and 12 h (Figure 9B). IFA indicated that compared with the 12 h infection ASFV group, the ASFV p30 protein fluorescence intensities of the BAY11-7082 groups were weaker than those of the ASFV group (Figure 9C).

## 4. Discussion

African swine fever virus (ASFV) is an exotic animal disease in China and was first reported in Kenya 100 years ago [30]. The first case of ASF in China was reported in August 2018, and quickly spread to all of China [19], causing huge economic losses. Presently, there are no effective vaccines or therapeutic drugs against ASF, making its prevention and control a global problem [31]; moreover, the immune response of animals to ASF is still unclear [32]. Additionally, apart from the development of effective vaccines, the development of antiviral drugs targeting the interaction between ASFV and host factors is an emerging prevention and control strategy [33]. Therefore, in-depth research on the immune response and pathogenic mechanisms of ASFV-infected hosts is of great significance for the development of effective ASF vaccines and antiviral drugs. A previous study reported that five tissues—lung, spleen, liver, kidney, and lymph nodes—synergistically responded to ASFV infection and resisted ASFV through inflammatory cytokine storms and interferon activation, through transcriptomic and proteomic analysis [34]. Based on RNA-Seq analysis, differential genes in acutely infected pigs, asymptomatically infected pigs, and healthy pigs were significantly enriched in immune pathways [35]. ASFV-Pig/HLJ/18 infection with PAMs transcriptomic data showed that ASFV significantly suppressed host immune responses and altered metabolism to promote viral replication and disease in pigs [36]. ASFV-CN/GS/2018 post-infection transcriptomic data showed that differential genes were significantly enriched in innate immune response, inflammatory response, and chemokine and apoptosis pathways, and the expression of some antiviral and inflammation-related factors also changed significantly. [37]. In this study, the transcriptomic and proteomic analysis of ASFV-infected PAMs found that ASFV-GZ201801 activated the NF-κB signaling pathway at the early stage of infection. In the present study, we demonstrated that the NF-κB inhibitor BAY11-7082 inhibited GZ201801 ASFV replication in PAMs, suggesting that drugs that inhibit the NF-κB signaling pathway could be applied in ASF treatment.

The ASFV primarily targets monocytes and macrophages, and these two cells play a key role in the host immune response [38]. However, research on the mechanism of interaction between ASFV and monocytes or macrophages is relatively comparative. ASFV infection triggers the upregulation of pro-inflammatory factors [39]. These pro-inflammatory factors contribute to the stimulation of protective immune responses, leading to the development of excessive systemic inflammatory reactions and inflammatory lesions [39]. In the present study, the results revealed that the NF-κB inhibitor BAY11-7082 significantly reversed ASFV-induced increase in the expression profiles of several pro-inflammatory factors, including IL-1β and IL-8 in PAMs. This finding suggested that BAT11-7082 modulated ASFV-induced inflammatory responses by decreasing the mRNA expression of pro-inflammatory cytokines in vitro.

NF-κB, a key transcription factor that regulates the activation of inflammatory cytokines, can be activated by viral infection, viral gene expression, or LPS stimulation [40], and it can be exploited by influenza viruses or type 1 HIV to sustain a high viral replication [41,42]. NF-κB is widely involved in inflammatory diseases, and several studies have focused on the development of anti-inflammatory drugs targeting the NF-κB signaling pathway [43]. Studies have shown that ASFV infection inhibits cell apoptosis in the early stages and activates apoptosis in the late stage [44]. This may be because ASFV activates the NF-κB signaling pathway in the early stage to inhibit cell apoptosis and maintain the level of virus replication in the host. During ASFV infection, the late inhibition of the NF-κB signaling pathway can promote cell apoptosis to enhance phagocytosis through apoptosis signals, increase virus spread, and cause serious pathological damage to the body. In this study, the NF-κB signaling pathway was activated in the early stage of ASFV infection and inhibited in the late stage, and this phenomenon can be confirmed in previous studies. Previous data have shown that the 32 kDa higher molecular mass form of A238L predominantly accumulated in the nucleus at later times post-infection, this form of the protein is co-precipitated with the NF-κB p65 subunit, suggesting that, particularly at late times p.i., A238L functions within the nucleus to inhibit NF-κB [45]. ASFV encodes a homologue of the inhibitor of apoptosis (IAP) that promotes cell survival by controlling the activity of the transcription factor NF-κB [46]. According to reports, ASFV IAP (A224L) is able to activate NF-κB, its effect being dependent on IKK activity [47]. This study uses the NF-κB pathway inhibitors BAY11-7082 to inhibit the expression of phospho-NF-κB p65 and p-IκB proteins and IL-1β and IL-8, thereby inhibiting the replication of ASFV.

In summary, the activation and regulation mechanism of NF-κB is a complex process, which is regulated by multiple mechanisms of the body and participates in the complex life process. When the body is stimulated by external stimulation to activate the NF-κB signaling pathway, it will secrete a variety of cytokines. The homeostasis of the body is disrupted, leading to the occurrence of various diseases. The results of this study demonstrate that ASFV replication can be inhibited in vitro by inhibiting the NF-κB signaling pathway and the production of IL-1β and IL-8 by the inhibitor BAY11-7082. These findings not only provide new insights into the related mechanism between the NF-κB signaling pathway and ASFV infection, but also indicate that the NF-κB pathway inhibitor BAY11-7082 is a potential immunomodulatory in the control of ASF. In recent years, the research on the related mechanism of the interaction between the virus and the NF-κB pathway has received extensive attention, and more studies have focused on the influence of the activation of the NF-κB pathway on the mechanism of viral infection. Therefore, studying the interaction between ASFV and the host NF-κB signaling pathway provides new ideas for preventing and controlling the outbreak and epidemic of ASF diseases.

## Figures and Tables

**Figure 1 viruses-14-00297-f001:**
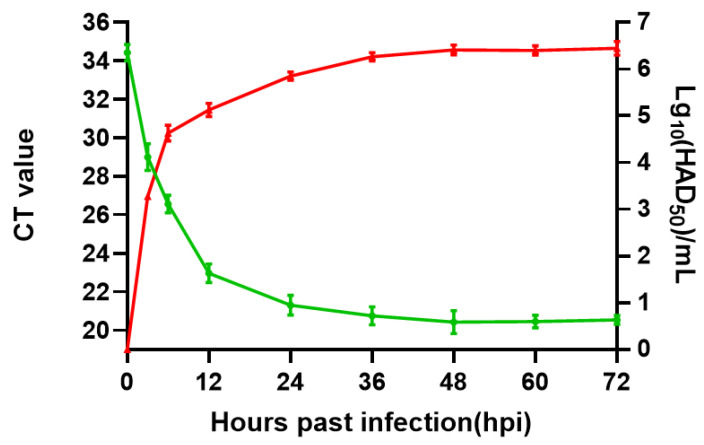
Proliferation and growth curves of GZ201801. Primary PAMs were infected with GZ201801 at an MOI of 1, and the supernatants and cells were mixed for infectious virus titter and viral p72 gene CT value by using HAD and qPCR assays, respectively.

**Figure 2 viruses-14-00297-f002:**
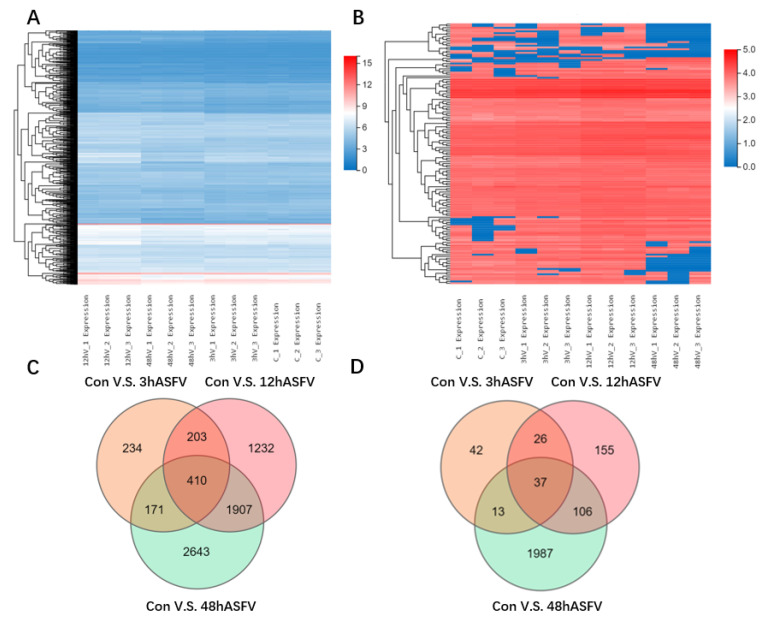
Conjoint analysis of transcriptomics and proteomics. Primary PAMs were infected with GZ201801 at an MOI of 1, and harvested samples 3 h, 12 h, and 48 h post-infection were sequenced for transcriptomics (**A**) and proteomics (**B**). Three sets of experiments were repeated at each time point, and the omics results were analyzed by cluster heat map. The differential genes (**C**) and differential proteins (**D**) at the three time points of ASFV infection were illustrated using Venn diagrams.

**Figure 3 viruses-14-00297-f003:**
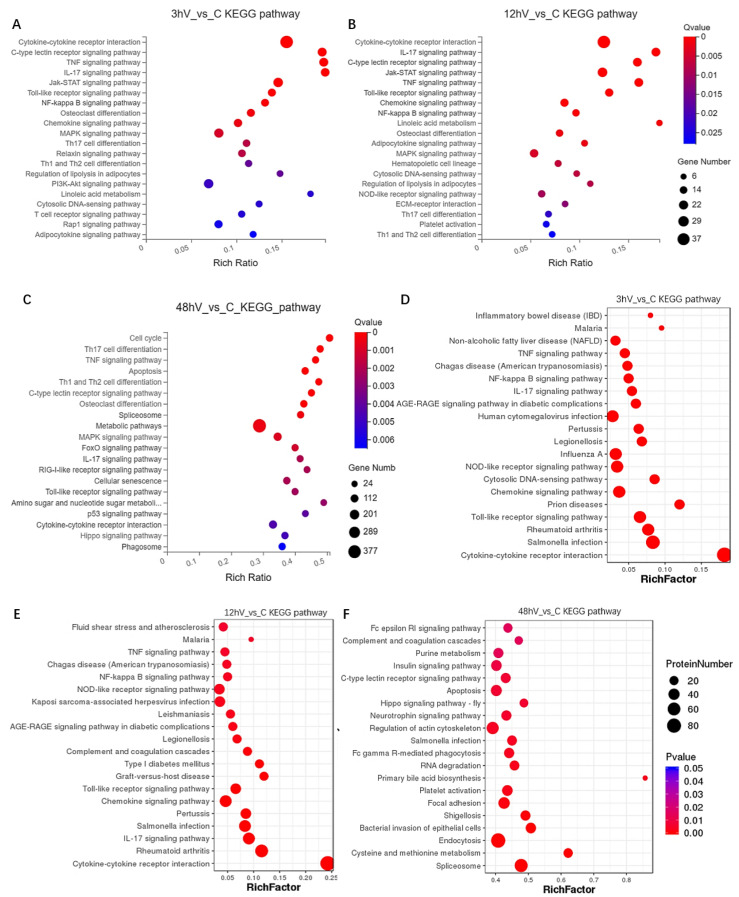
KEGG pathway enrichment analysis of DEGs and DEPs. Differentially expressed genes (**A**–**C**) and differentially expressed proteins (**D**–**F**) at the three time points of ASFV infection were annotated using KEGG pathway enrichment analysis.

**Figure 4 viruses-14-00297-f004:**
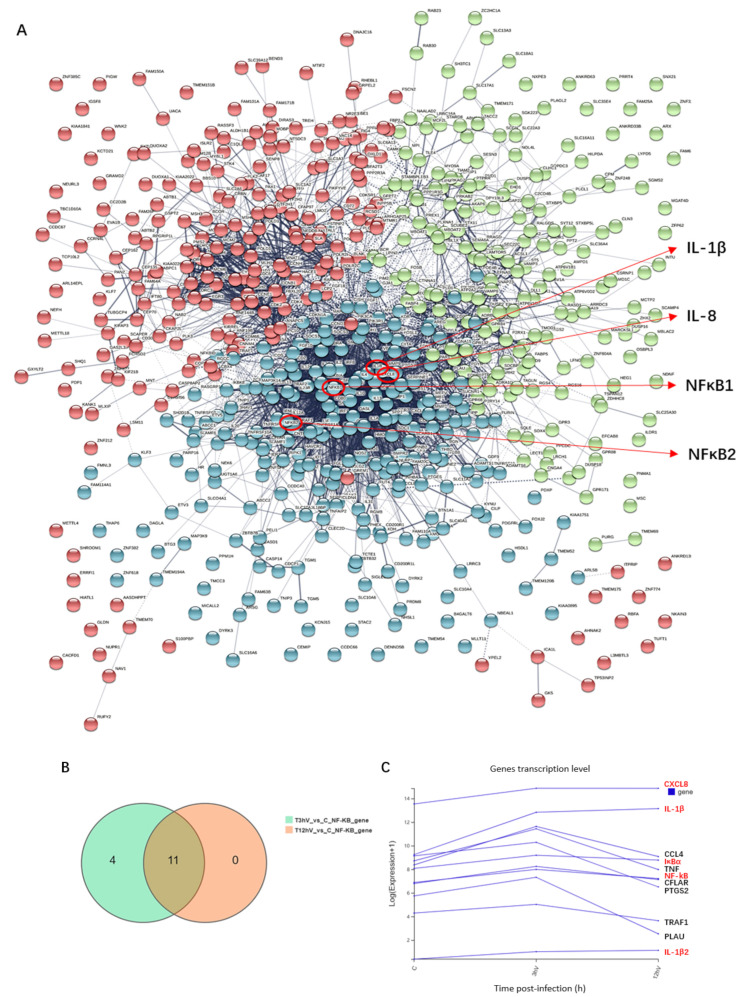
NF-κB signaling pathway analysis after ASFV infection. (**A**) Analysis of protein network interaction of differential genes. (**B**) Venn diagram of differentially expressed genes in the NF-κB signaling pathway after 3 h and 12 h infection with ASFV. (**C**) Variation in the trend of differentially expressed genes in the NF-κB signaling pathway with an increase in ASFV infection duration.

**Figure 5 viruses-14-00297-f005:**
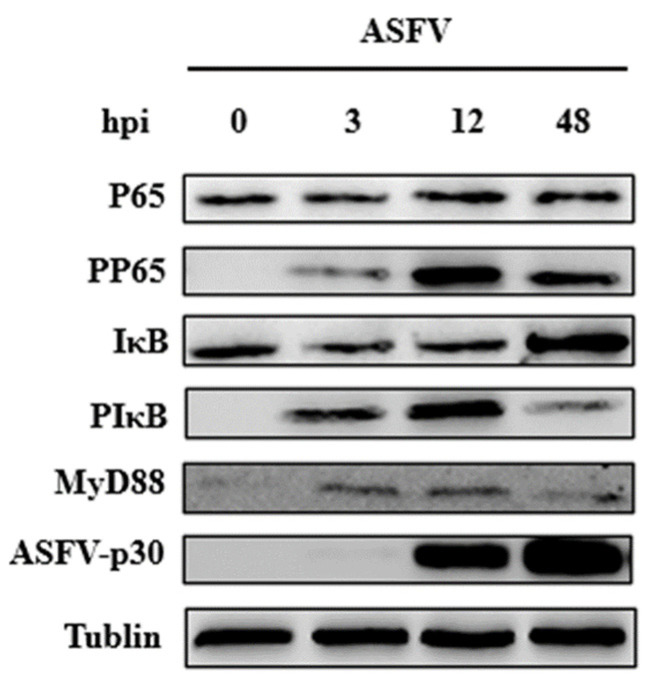
The expression of proteins in PAMs infected by ASFV. Variation in the trend of P65, PP65, IκB, PIκB, MyD88, and p30 proteins after 0, 3, 6, 12, and 48 h infection with ASFV duration by Western Blotting.

**Figure 6 viruses-14-00297-f006:**
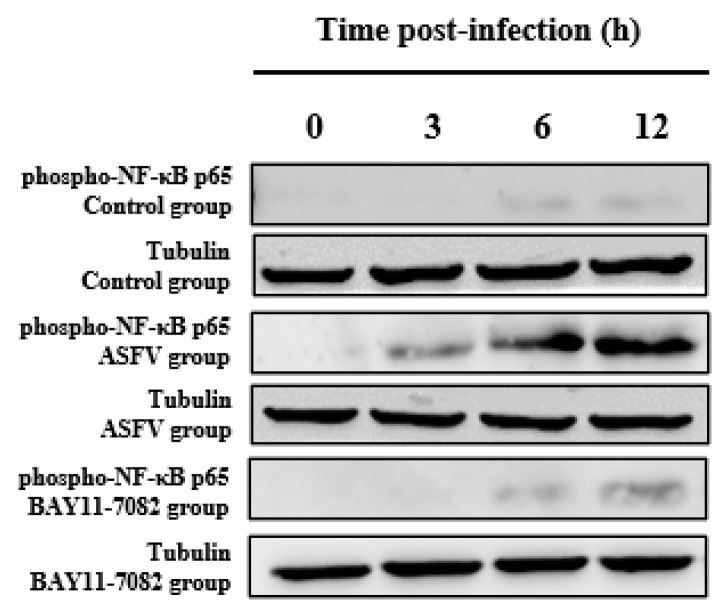
Effect of BAY11-7082 on the phosphor-NF-κB p65 protein after ASFV infection. Western blotting was used to measure the expression of the phosphor-NF-κB p65 protein at 0, 3, 6, and 12 h in each group of PAMs. Levels of pp65 increased after ASFV infection; however, this effect was reversed by the NF-κB inhibitor BAY11-7082. The expression of tubulin was used as a positive control.

**Figure 7 viruses-14-00297-f007:**
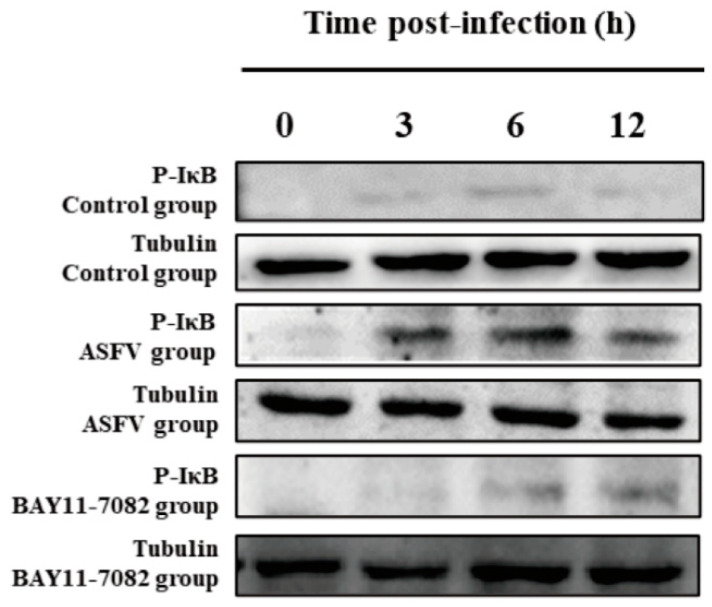
Effect of BAY11-7082 on p-IκB protein after ASFV infection. Western blotting was used to measure the expression of the p-IκB protein in the cytoplasm at 3, 6, and 12 h in each group of PAMs. Levels of p-IκB increased after ASFV infection relative to controls, and this effect was abrogated by the NF-κB inhibitor BAY11-7082. Tubulin expression was used as a positive control.

**Figure 8 viruses-14-00297-f008:**
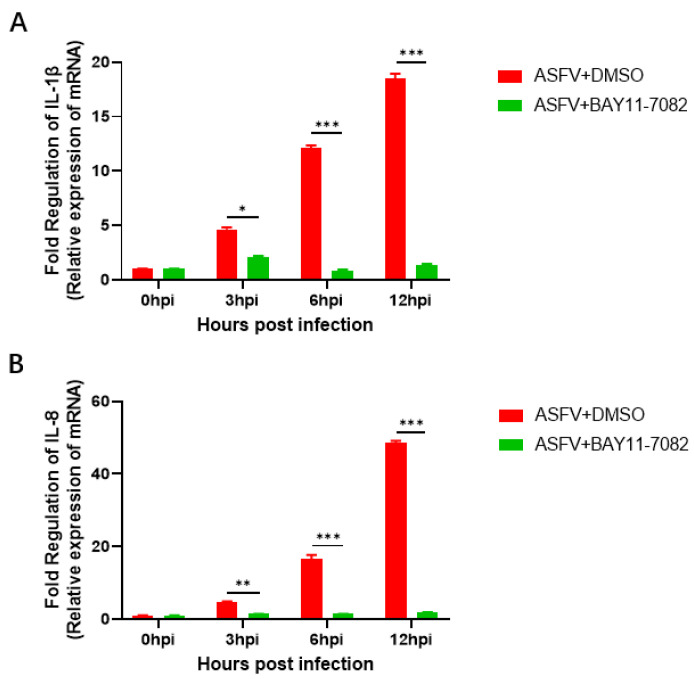
Changes in the expression of IL-1β and IL-8 at the mRNA level. Real-time PCR was used to assess the expression of IL-1β and IL-8 at the mRNA level in PAMs infected with ASFV, BAY11-7082 after 0, 3, 6 and 12 h. (**A**) Increased expression of IL-1β was observed in ASFV in-fected cells, and this was reversed by BAY11-7082. (**B**) Increased expression of IL-8 was observed in ASFV infected cells, and this was reversed by BAY11-7082. Each datum represents results of three independent experiments (means ± SD). Significant differences compared with the control group are denoted by * (*p* < 0.05), ** (*p* < 0.01) and *** (*p* < 0.001).

**Figure 9 viruses-14-00297-f009:**
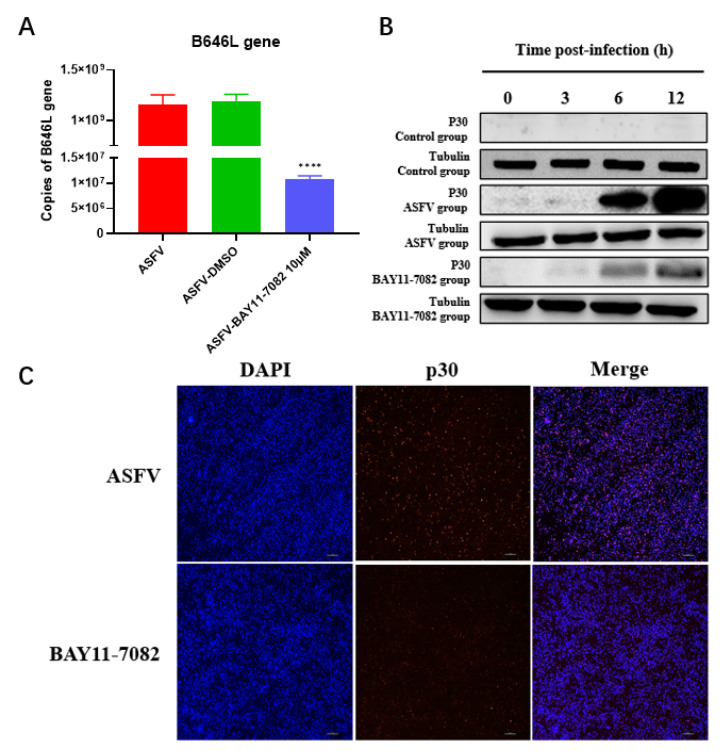
Antiviral activity of BAY11-7082 against ASFV. (**A**) PAMs infected with ASFV (1 MOI) at 37 °C and then cultured in fresh medium supplemented with 10 μM BAY11-7082. The expression levels of ASFV p72 in PAMs were detected by real-time PCR analysis at 12 h. Each data represents results of three independent experiments (means ± SD). (**B**) ASFV p30 protein expression levels in cells treated with BAY11-7082 were evaluated by Western blotting. (**C**) Antiviral activity of BAY11-7082 against ASFV was determined in PAMs by IFA. PAMs were seeded in 24-well plates and infected with ASFV (1 MOI) at 12 h, and then incubated with RPMI 1640 supplemented with the indicated concentration of BAY11-7082. The p30 protein was used as an indicator of ASFV infection, and IFA was performed using mouse anti-p30 protein antibody and goat anti-mouse IgG Alexa Fluor. Nuclei were counterstained with DAPI (blue). Each datum represents results of three independent experiments (means ± SD). The images above represent three independent IFA trials with similar results. Significant differences compared with the control group are denoted by **** (*p* < 0.001).

**Table 1 viruses-14-00297-t001:** Primer sequences used in this study for PCR and real-time PCR in pigs.

Gene	Primer Sequence (5′-3′)
CADC-B646L-rPCRF	ATAGAGATACAGCTCTTCCAG
CADC-B646L-rPCRR	GTATGTAAGAGCTGCAGAAC
CADC-B646L-Probe	FAM-TATCGATAAGATTGAT-MGB
B646L-F	TGAAATAAAATGGAAGCCCACAGATC
B646L-R	ACACTGTACAACATTGCGTAAAAGC
GAPDH-F	GCAAAGACTGAACCCACTAATT
GAPDH-R	TTGCCTCTGTTGTTACTTGGAG
IL-1β-F	ACCTGGACCTTGGTTCTCTG
IL-1β-R	CATCTGCCTGATGCTCTTG
IL-8-F	CACTGTGAAAATTCAGAAATCATTGT
IL-8-R	CTTCACAAATACCTGCACAACC

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
