# Peer review of "Effects of the NF-κB Signaling Pathway Inhibitor BAY11-7082 in the Replication of ASFV"

_viruses, 2022, doi:10.3390/v14020297_

Round 1

Reviewer 1 Report

Thanks to the authors for their answers and modifications in the manuscript regarding to my initial concerns. I think the results of this study are in accordance with others that express the potential role of proinflammatory cytokines in the pathogenesis of ASFV (Zhu et al., 2019, Wang et al., 2021).  At this point, I consider that this new version was significantly improved. I don’t have additional concerns to recommend this study for publication.

Author Response

Dear Reviewer, thank you for taking the time to review this manuscript.

Reviewer 2 Report

The manuscript entitled “Effects of the NF-kB signalling pathway inhibitor BAY 11-7082 in the replication of ASFV” describes a transcriptomic and proteomic analysis of cells infected with ASFV, and testing the efficiency of known NF-kB inhibitor. Since ASFV poses a serious concern for worldwide pig industry and food safety, the subject is of particular importance in the context of pig immune response and vaccine development. Nevertheless, the aim of the study is not clearly stated: multiomics analysis of ASFV infected host cells or antiviral effect of BAY11-7082 inhibitor and the mechanism of its action?

Similar transcriptomic analyses were recently published other authors: (Fan et al., 2021; Ju et al., 2021; Sun et al., 2021; Yang et al., 2021). English language needs to be carefully checked since it contains various minor and major errors. The overall content seems to be chaotic and sometimes hard to understand, however, the manuscript is interesting and contains valuable information and may be recommended for publication after major revision.

General comments:

Aim: aim of the study is not clearly stated, why did authors perform very complex and expensive  multiomics analyses just to show that the inhibitor acts on the NF-kB pathway? On the other hand, if they decided to do so, the huge amount of generated data may be used to show how complex is the interaction between ASFV and host cell. For many years it has been widely known that NF-kB pathway is inhibited by ASFV infection, leading to the host immune response evasion. Contrary to previous data, the results obtained by the authors suggests the increased transcription of proteins related to this pathway, which needs deeper insight and discussion since Ju et al. showed that some certain pathways are variably expressed dependently on the time post infection. Please compare also Yang et al. which showed that ASFV infection upregulates antiviral and anti-inflammatory factors.   

Language: some sentences are obscure and hard to understand, thus the language needs to be improved. Abstract and introduction must be improved. Introduction lacks the information that ASFV infects only pigs, but on the other hand it contains unnecessary, detailed information regarding ASFV virion architecture. The sentence about transmission by insects may lead to false conclusion that it is the only transmission mode, whereas the information regarding direct transmission between infected animals or their carcasses is omitted.

Methodology: lacks important information to replicate the experiment by other authors, especially unit 2.3 and 2.6, 2.7 are obscure. Please add more precise information what software was used for the analysis and graphic presentation of the results, how the raw reads were filtered, etc.

Line 142: what kit was used to do so?

Line 144-145: RNA purification kit was used to cDNA purification, repair base ends and connect the adapters? This sentence sounds at least confusing.

Line 148: which Illumina platform?

Line 154: what reference sequence?

Results:

Figure 1: caption need to be improved, lack of general information regarding what is presented at the figure

Line 264-266: Why these specific proteins were selected? Their function should be explained here. What is MyD88? What is the role of the phosphorylation?  The whole sentence is chaotic and the final conclusion is not clear for the audience unfamiliar with cell biology and immunology.

Figure 4:  There is no D, please improve the caption. 4C – axes captions should be more precise. The changes between time 0, 3 and 12 hpi in most cases does not seem to be significant. Elaborate. What means the red colour in the gene names?

Line 279: again, why this protein and what is it role?

Line 309: inoculated =/= infected.

Line 315, 323: RT-PCR does not mean real-time PCR. Please improve.

Section 3.7: Why B646L gene was used for qPCR, whereas western blot analyses were performed using other protein? Lack of consistency. The application of logarithmic reduction value (LRV) instead of raw logarithms would ease the understanding of the figure. What is exact reduction of virus titre in the inhibitor group? Something about 2 logs measured by real-time PCR. Consider the confirmation of the results by virus titration in cell culture.

Discussion must relate to the results obtained by (Fan et al., 2021; Ju et al., 2021; Sun et al., 2021; Yang et al., 2021).

Line 360: what is the level on inhibition? Is it complete? Maybe it would be worth to test other inhibitor concentrations and calculate IC50 ?

Line 382-383: “the late inhibition of the NF-κB signaling pathway can promote cell apoptosis to enhance phagocytosis through apoptosis signals, increase virus spread, and cause serious pathological damage” – this thesis is very interesting and stand in line with previous data, but is contrary to your study. Definitely, it is worth to prove this hypothesis.

Line 403-404 : This sentence needs to be improved.

References:

Fan, W., Cao, Y., Jiao, P., Yu, P., Zhang, H., Chen, T., Zhou, X., Qi, Y., Sun, L., Liu, D., Zhu, H., Liu, W., Hu, R., & Li, J. (2021). Synergistic effect of the responses of different tissues against African swine fever virus. Transboundary and Emerging Diseases, August, 1–12. https://doi.org/10.1111/tbed.14283

Ju, X., Li, F., Li, J., Wu, C., Xiang, G., Zhao, X., Nan, Y., Zhao, D., & Ding, Q. (2021). Genome-wide transcriptomic analysis of highly virulent African swine fever virus infection reveals complex and unique virus host interaction. Veterinary Microbiology, 261(August), 109211. https://doi.org/10.1016/j.vetmic.2021.109211

Sun, H., Niu, Q., Yang, J., Zhao, Y., Tian, Z., Fan, J., Zhang, Z., Wang, Y., Geng, S., Zhang, Y., Guan, G., Williams, D. T., Luo, J., Yin, H., & Liu, Z. (2021). Transcriptome Profiling Reveals Features of Immune Response and Metabolism of Acutely Infected, Dead and Asymptomatic Infection of African Swine Fever Virus in Pigs. Frontiers in Immunology, 12(December), 1–15. https://doi.org/10.3389/fimmu.2021.808545

Yang, B., Shen, C., Zhang, D., Zhang, T., Shi, X., Yang, J., Hao, Y., Zhao, D., Cui, H., Yuan, X., Chen, X., Zhang, K., Zheng, H., & Liu, X. (2021). Mechanism of interaction between virus and host is inferred from the changes of gene expression in macrophages infected with African swine fever virus CN/GS/2018 strain. Virology Journal, 18(1), 1–16. https://doi.org/10.1186/s12985-021-01637-6

Author Response

Hello dear teacher, thank you for your valuable comments, I have responded to your questions one by one. Details are as follows:

  1. Changed "transmission by insects" to "transmitted by the domestic pig, wild boars, and insect vectors", and has been marked yellow. (Line35-37)

African swine fever (ASF) is an acute and highly infectious disease of pigs caused by the African swine fever virus (ASFV) and is transmitted by domestic pig, wild boars, and insect vectors [1].

  1. The content of the ASFV virion structure in the introduction has been omitted, and the clinical symptoms of ASFV-infected animals have been supplemented and have been marked yellow. (Line43-49)

Acute infection of pigs with ASFV can result in up to 100% mortality. Currently, there are no commercially approved vaccines or antiviral drugs to control the disease, so control measures rely on culling infected animals, restricting animal movement, and Biosecurity prevention and control [5, 6]. Highly virulent isolates of ASFV often cause a peracute to acute disease progression with high fever (>41 °C) and a range of clinical signs, including anorexia and lethargy, which occur within a few days of infection [7].

  1. The HAD50 experimental and computational methods in 2.3 have been supplemented and more accurate information added and has been marked yellow. (Line122-127)

As previously described [28], primary PAMs were cultured in 96-well plates and infected with 10-fold diluted ASFV (GZ201801). The quantity of ASFV was determined by the identification of characteristic rosette formation representing hemadsorption of erythrocytes around infected cells. Cultures were observed for HAD phenomena over 7 days., and HAD50 was calculated using the method of Reed and Muench [29]. Primary PAMs were infected with GZ201801-ASFV at an MOI of 1.

  1. Software information and methods for filtering raw data have been added in 2.6 and have been marked yellow. (Line144-157)

Total RNA was processed by mRNA enrichment method, mRNA enrichment: mRNA with polyA tail was enriched with magnetic beads with OligodT. The obtained RNA was fragmented by interrupting buffer, reverse transcribed with random N6 primers, and cDNA double-stranded was synthesized to form double-stranded DNA. The end of the synthesized double-stranded DNA is blunted and phosphorylated at the 5' end, forming a sticky end with an "A" protruding from the 3' end, and then ligated with a bubbling linker with a protruding "T" at the 3' end. The ligated products are amplified by PCR with specific primers. The PCR product was heat-denatured into single-stranded DNA, and then a single-stranded circular DNA library was obtained by circularizing the single-stranded DNA with a bridge primer. The constructed library is checked for quality and sequenced after it is qualified. This project uses the DNBSEQ platform to sequence the samples, and each sample produces an average of 6.68G of data. The average alignment rate of the sample alignment genome was 90.86%, and the average alignment rate of the aligned gene set was 66.68%; a total of 17,158 genes were detected.

  1. Software for analyzing and presenting plots of results has been added in 2.7 and detailed information has been added, and has been marked yellow. (Line159-163, 166-168)

  1. Question: Line 142: what kit was used to do so?

Answer: This operation is carried out through the kit independently developed by BGI.

  1. Question: Line 144-145: RNA purification kit was used to cDNA purification, repair base ends, and connect the adapters? This sentence sounds at least confusing.

Answer: The sentence has been edited, and has been marked yellow. (Line145-147)

The obtained RNA was fragmented by interrupting buffer, reverse transcribed with random N6 primers, and cDNA double-stranded was synthesized to form double-stranded DNA.

  1. Question: Line 148: which Illumina platform?

Answer: The content has been modified and supplemented, and has been marked yellow. (Line154-155)

This project uses the DNBSEQ platform to sequence the samples, and each sample produces an average of 6.68G of data.

  1. Question: Line 154: what reference sequence?

Answer: Reference sequence has been added, and has been marked yellow. (Line166-168)

After quality control, clean reads were aligned with the reference sequence GCF_000003025.6_Sscrofa11.1(NCBI).

  1. Question: Figure 1: caption needs to be improved, lack of general information regarding what is presented at the figure

Answer: The title has been revised and added, and has been marked yellow. (Line228)

Figure 1. Proliferation and growth curves of GZ201801.

  1. Question: Line 264-266: Why these specific proteins were selected? Their function should be explained here. What is MyD88? What is the role of phosphorylation? The whole sentence is chaotic and the final conclusion is not clear for the audience unfamiliar with cell biology and immunology.

Answer: Explain why these proteins were selected and their functions, describe the role of MyD88 and protein phosphorylation and modify and supplement the statement, which has been marked yellow. (Line273-278, 280-287)

Venn diagram analysis showed that among the common DGEs in the 3h and 12h infection groups compared with the control group, 11 differential genes were enriched in the NF-κB signaling pathway, including CXCL8, IL-1β, CCL4, IκBα, TNF, NF-κB, CFLAR, PTGS2, TRAF1, PLAU and IL-1β2 (Fig. 4B). There was an increase in the RNA levels of IL-1β, IL-1β2, IL-8, NF-κB, and IκBα with up-regulating in infection time and shown in red font. (Fig. 4C).

When the NF-κB signaling pathway is activated, both NF-κB and IκB proteins with a dimer structure undergo phosphorylation modification, and the phosphorylated IκB (PIκB) is degraded, resulting in the phosphorylated NF-κB (PP65) entering the nucleus and causing an immune response. And myeloiddifferentiationfactor88 (MyD88) is a key linker molecule that activates the NF-κB signaling pathway. The results show that the levels of phospho-NF-κB p65, p-IκB, and MyD88 proteins increased with infection time from 3 h–12 h post-infection and decreased after 48 h post-infection (Fig. 5), indicating the MyD88/NF-κB signaling pathway is activated early in ASFV infection.

  1. Question: Figure 4: There is no D, please improve the caption. 4C – axes captions should be more precise. The changes between time 0, 3, and 12 hpi in most cases do not seem to be significant. Elaborate. Does that mean the red color in the gene names?

Answer: Due to a mistake, the D in the legend has been deleted, and the content of the coordinate axis of Figure 4C has been improved.

4C is only the result of the omics analysis. We can see the temporal trend of differential genes enriched in the NF-κB signaling pathway from the figure. Therefore, according to this trend, we conducted subsequent ASFV infection to regulate the host NFκB signaling pathway. The expression levels of key proteins in this pathway were detected.

Gene names in red represent genes whose transcription level is up-regulated with infection time, this explanation has been added in the manuscript and has been marked yellow. (Line277-278)

There was an increase in the RNA levels of IL-1β, IL-1β2, IL-8, NF-κB, and IκBα with up-regulate in infection time and shown in red font. (Fig. 4C).

  1. Question: Line 279: again, why this protein, and what is its role?

Answer: This has been explained in the manuscript, and has been marked yellow. (Line280-287)

When the NF-κB signaling pathway is activated, both NF-κB and IκB proteins with a dimer structure undergo phosphorylation modification, and the phosphorylated IκB (PIκB) is degraded, resulting in the phosphorylated NF-κB (PP65) entering the nucleus and causing an immune response. And myeloiddifferentiationfactor88 (MyD88) is a key linker molecule that activates the NF-κB signaling pathway. The results show that the levels of phospho-NF-κB p65, p-IκB, and MyD88 proteins increased with infection time from 3 h–12 h post-infection and decreased after 48 h post-infection (Fig. 5), indicating the MyD88/NF-κB signaling pathway is activated early in ASFV infection.

  1. Question: Line 309: inoculated =/= infected.

Answer: It has been modified, and has been marked yellow. (Line334)

  1. Question: Line 315, 323: RT-PCR does not mean real-time PCR. Please improve.

Answer: It has been modified, and has been marked yellow. (Line333, 341, 356)

  1. Question: Section 3.7: Why B646L gene was used for qPCR, whereas western blot analyses were performed using other proteins? Lack of consistency. The application of logarithmic reduction value (LRV) instead of raw logarithms would ease the understanding of the figure. What is the exact reduction of virus titer in the inhibitor group? Something about 2 logs measured by real-time PCR. Consider the confirmation of the results by virus titration in cell culture.

Answer: B646L gene is the ASF detection primer recommended by OIE and CADC, so B646L was selected as the detection gene of qPCR in this experiment. In this experiment, the NF-κB signaling pathway was activated in the early stage of ASFV infection (3h and 12h), so the early ASFV protein p30 was selected for detection.

The ordinate axis of 9A has been modified.

A histogram of the detection of viral titers with the inhibitor BAY11-7082 has been added and placed in Supplementary Material 1, and has been marked yellow. (Line342-344, 433)

  1. Question: Discussion must relate to the results obtained by (Fan et al., 2021; Ju et al., 2021; Sun et al., 2021; Yang et al., 2021).

Answer: These four articles have been discussed and supplemented in this manuscript., and has been marked yellow. (Line376-387)

A previous study reported that five tissues, lung, spleen, liver, kidney, and lymph nodes, synergistically responded to ASFV infection and resisted ASFV through inflammatory cytokine storm and interferon activation through transcriptomic and proteomic analysis [34]. Based on RNA-Seq analysis, differential genes in acutely infected pigs, asymptomatically infected pigs, and healthy pigs were significantly enriched in immune pathways [35]. ASFV-Pig/HLJ/18 infection with PAMs transcriptomic data showed that ASFV significantly suppressed host immune responses and altered metabolism to promote viral replication and disease in pigs [36]. ASFV-CN/GS/2018 post-infection transcriptomic data showed that differential genes were significantly enriched in innate immune response, inflammatory response, chemokine and apoptosis pathways, and the expression of some antiviral and inflammation-related factors also occurred significant changes [37].

  1. Question: Line 360: what is the level of inhibition? Is it complete? Maybe it would be worth testing other inhibitor concentrations and calculating IC50?

Answer: In this experiment, BAY11-7082 plays a role in inhibiting the NF-κB signaling pathway. The activity detection (CCK8) of BAY11-7082 after treatment of PAMs has been supplemented in this manuscript, and the figure has been added in Supplementary Material 2 and has been marked yellow. (Line301-302, 451)

From this figure, it was found that when PAMs were treated with BAY11-7082 at a concentration of 10 μM, the cell activity was 80%. When the concentration was too large, PAMs would be damaged. Therefore, we selected 10 μM BAY11-7802 for NF-κB pathway activation. At the same time, it was found that this concentration can significantly inhibit the proliferation of ASFV, so we finally selected a concentration of 10 μM for the experiment.

  1. Question: Line 382-383: “the late inhibition of the NF-κB signaling pathway can promote cell apoptosis to enhance phagocytosis through apoptosis signals, increase virus spread, and cause serious pathological damage” – this thesis is very interesting and stand in line with previous data, but is contrary to your study. Definitely, it is worth proving this hypothesis.

Answer: The results of this experimental study are that ASFV infection activates the NF-κB signaling pathway in the early stage and inhibits the NF-κB signaling pathway in the late stage. And while studies have shown that early inhibition of the NF-κB pathway can inhibit viral proliferation, what is written in the Discussion in this manuscript is that late inhibition of the NF-κB pathway by ASFV activates apoptosis and promotes viral spread. (Line407-416)

  1. Question: Line 403-404: This sentence needs to be improved.

Answer: This sentence has been improved and perfected, and has been marked yellow. (Line438-440)

Therefore, studying the interaction between ASFV and the host NF-κB signaling pathway provides new ideas for preventing and controlling the outbreak and epidemic of ASF diseases.

Round 2

Reviewer 2 Report

Thank Your for improvement of the indicated issues. The language should be improved by the English native speaker since it contains minor errors. 

This manuscript is a resubmission of an earlier submission. The following is a list of the peer review reports and author responses from that submission.

Round 1

Reviewer 1 Report

Gao et al report on porcine alveolar macrophage (PAM) responses to African Swine Fever Virus (ASFV) infection. Using transcriptomic and proteomic analyses at 3, 12 and 48 hours post-infection (hpi), differentially expressed products were identified, including genes and proteins of the NF-kB pathway and cytokines such as TNF and IL-1. Using Western blotting, viral infection was found to induce phosphorylation of NF-kB p65 and IkB, and cytokine expression, indicative of NF-kB activation, while treatment of PAMs with NF-kB inhibitor BAY11-7082 previous to virus infection prevented these changes and inhibited ASFV replication. Overall, the experiments are well done. However, the interest of the paper is limited as some of the research here has been described before. For example, induction of TNFα by PAMs infected with ASFV has been described by Gomez del Moral et al, 1999. Cytokine (IL-1, TNF) induction by ASFV infection was reviewed by Gomez-Villamandos, 2013. Activation of NF-kB by ASFV IAP (A224L) has been reported, etc. There are also issues regarding clarity and presentation that the authors should address.

Major Comments

- Fig 2. Most of the text in the figure is unreadable because of the font size. Figures A and B, some information is required in the legend for interpretation of these figures. Also, no control was included. Are the three time points represented in A and B? Clarify.

C and D, two circles are labeled 3 hs and no 48 h circles are shown. I understand this is a printing mistake.

Panels E-J, again, as shown they are useless because of the font size and lack of explanation in the text/legend.

- L252, “indicating an increase in the translocation of NF-kB…” This is incorrect because p65 nuclear translocation experiments were not performed here. Thus “suggesting an increase in the translocation, etc” or “suggesting NF-kB activation” might be appropriate.

- Fig 7A. to my knowledge this is the first study showing NF-kB positively impacting ASFV replication, so it will be important to provide further evidence for this effect. 1- What statistical analysis was used for comparison and what p values were obtained? 2- Have the authors attempted to repeat the experiments using a different NF-kB inhibitor? A direct effect of BAY11-7082 on one or more aspects of ASFV replication cannot be ruled out; 3- It is not clear at what time post-infection is the medium with inhibitor added to the PAMs, please clarify. Have the authors attempted to repeat the experiment with PAMs pre-treated with inhibitor?

L313, “ at the indicated time points”, no time points are indicated, rather there seems to be a single time point in Fig 7A. Indeed, running the experiment at different time points would be ideal. Clarify.

- Fig 7B and 7C convey basically the same information.

L361/362, the statement is not supported by the data, neither IkB degradation nor NF-kB translocation were investigated here.

- Discussion. Surprisingly, previous work regarding ASFV and NF-kB has not been addressed by the authors. ASFV has been shown to stimulate (via A224L) or inhibit (via A238L) NF-kB at different times post-infection. The authors should discuss their results in face of these studies.

Minor comments

- L1, Title: Inhibition of African swine fever virus replication by….

- L95, PAMs preparation should be described or otherwise referenced.

- L114, likely “10-fold diluted” can be removed here as long as the used MOI is 1.

- L248, add (Fig. 3D) at the end of sentence.

- Fig. 3A and B. The same clarity issues as in Fig 2 apply here. In the legend, statement in L257 is not very descriptive. “Expression levels of differentially expressed genes at 0, 3 and 12 pi…etc”.

- L265. Same issue as in L252, p65 nuclear translocation experiments were not performed.

Tubulin as positive control? Tubulin is not  positive control in this experiment.

- Figs 4, 5 and 7B, top, should read Time post-infection (h).

- L284, PAMs instead of MAPs.

- Results in Fig 6 are expected since the expression of these genes is upregulated by NF-kB.

- Fig 7B. The antibody against p30 is not described in Material and Methods. Is the p30 antibody used for IF the same used in Western blot?

- L333, Inhibited better describe the effect than suppressed.

- L339, #34 is likely not the right reference for this sentence. Check.

Reviewer 2 Report

I reviewed the manuscript entitled “Inhibit the Replication of African Swine Fever Virus by Inhibiting the NF-κB Signaling Pathway and Interleukin-1 beta and Interleukin-8 Production”.  In this study, using the NF-κB inhibitor BAY11-7082, authors aimed to evaluate effect of NF-κB during the replication in-vitro ASFV.

Overall, the authors show some evidence about the positive role of NF-κB during the replication of ASFV. However, I consider that the study design has several flaws that need to be addressed to support the conclusions of this study.

  1. English proofreading should be applied to the entire text. There are several sentences where the ideas are not clear, and it makes hard to judge the exact meaning
  2. I found the title very speculative, particularly because the authors don’t show a real evidence about the specific role of Interleukin-1 beta and Interleukin-8 in the replication of ASFV. I would suggest changing the title for something like: Effects of the NF-κB inhibitor BAY11-7082 in the replication of ASFV.
  3. The introduction section should be restructured to include more information regarding the role of NF-κB pathway (positive and negative) in the replication of viruses and highlight what is known in ASFV. It makes help to support in a better way the aim of the study. Below you will find some references in ASFV all of them absent in the reference section of this study (see below). Also, I would suggest including more information about BAY11-7082(There are plenty of information already published). In the abstract authors present a statement that looks like they are discovering the function of this drug.
  4. Although the authors show in figure one the correlation between qPCR CT values and log10 HAD50/mL, I consider that to properly evaluate the effect of BAY11-7082 the results should be presented as log10 HAD50/mL to detect the presence of infectious virus. My main concern about using the qPCR is that the inhibition may be happening just for certain viral proteins that might be harboring NF-κB binding promoters (Santoro et al., 2003). Including both approaches and a set of different viral genes evaluated by qPCR may give the results a better perspective.
  5. I consider that experiments to evaluate the effect of BAY11-7082 should be conducted using different doses and different time points of collection. I think is important to see if viral yields are low just during the first hours of the infection or it is maintained for the long term.
  6. Discussion should be enriched. Please, discuss information about A238L protein of ASFV that prevents the nuclear translocation of NF-κB. Does ASFV need NF-κB just for a specific step in the replication? Other hypothesis about the positive role of NF-κB during the replication of ASFV may be the suppression of apoptosis that would increase the viral survival.

References

African swine fever virus A238L inhibitor of NF-kappaB and of calcineurin phosphatase is imported actively into the nucleus and exported by a CRM1-mediated pathway

Rhiannon N Silk 1, Gavin C Bowick 1, Charles C Abrams 1, Linda K Dixon 1

African Swine Fever Virus IAP-Like Protein Induces the Activation of Nuclear Factor Kappa B

Clara I. Rodríguez, María L. Nogal, Angel L. Carrascosa, María L. Salas, Manuel Fresno, and Yolanda Revilla*

African Swine Fever Virus Ubiquitin-Conjugating Enzyme Is an Immunomodulator Targeting NF-κB Activation

Lucía Barrado-Gil 1, Ana Del Puerto 1, Inmaculada Galindo 1, Miguel Ángel Cuesta-Geijo 1, Isabel García-Dorival 1, Carlos Maluquer de Motes 2, Covadonga Alonso 1